# AGS-SSD: Attention-Guided Sampling for 3D Single-Stage Detector

**Hanxiang Qian, Peng Wu, Bei Sun and Shaojing Su \***

College of Intelligence Science and Technology, National University of Defense Technology, Changsha 410073, China; qianhanxiang_@nudt.edu.cn (H.Q.); pengwu@nudt.edu.cn (P.W.); sunbei08@nudt.edu.cn (B.S.)
\* Correspondence: ssjing@nudt.edu.cn

**Abstract:** 3D object detection based on LiDAR point cloud has always been challenging. Existing point cloud downsampling approaches often use heuristic algorithms such as farthest point sampling (FPS) to extract the features from a massive raw point cloud. However, FPS has disadvantages such as low operating efficiency and inability to sample key areas. This paper presents an attention-guided downsampling method for point-cloud-based 3D object detection, named AGS-SSD. The method contains two modules: PEA (point external attention) and A-FPS (attention-guided FPS). PEA explores the correlation between the data and uses the external attention mechanism to extract the semantic features in the set abstraction stage. The semantic information, including the relationship between the samples, is sent to the candidate point generation module as context points. A-FPS weighs the point cloud according to the generated attention map and samples the foreground points with rich semantic information as candidate points. The experimental results show that our method achieves significant improvements with novel architectures against the baseline and runs at 24 frames per second for inference.

**Keywords:** 3D single-stage object detection; point downsampling; external attention

## 1. Introduction

With the development of society and growth of the economy, the market share of vehicles powered by renewable energy is steadily increasing owing to their advantages of environmental friendliness and high efficiency. In addition, the autonomous driving technology is gradually maturing. The environment perception system is a core component of autonomous driving and is responsible for vehicle planning and motion. Improvements in deep learning and computing power have enabled environmental perception technology to solve complex recognition and perception problems that cannot be solved via traditional methods. An important task for which the environmental perception system is responsible is 3D object detection.

Three-dimensional (3D) object detection entails identifying the category of the object in 3D space and marking it with the smallest 3D bounding box. Compared with 2D object detection, the representation of the 3D bounding box has three additional attitude angles and one additional dimension of position and size. In this regard, LiDAR can provide high precision, high adaptability, continuous detection and tracking capabilities under unfavourable lighting conditions such as that obtained with a backlight at night, and the collected laser point cloud data have accurate depth information and obvious three-dimensional spatial characteristics. Point cloud data have gradually become indispensable for perception in autonomous driving. The point cloud 3D object detection algorithm obtains the spatial information in the point cloud data for detecting vehicles, pedestrians, objects and other targets in the autonomous driving scene. The detection results provide path planning information and warn against potential threats to enable the autonomous driving vehicle to drive smoothly and safely [1].

Because laser point cloud data are characterised by sparseness, unstructuredness and massiveness, the direct application of a convolutional neural network [2,3], as is commonly used in 2D image processing to process point cloud data, is problematic in that it is computationally intensive and produces poor results. Therefore, the development of an efficient method to process point clouds for 3D object detection is an essential research topic.

Owing to the unstructured and disordered nature of point clouds, early studies adopted the method of converting 3D point clouds into 2D images [4–6], for example, transforming the 3D point clouds from a bird's eye view or multi-view and then using mature 2D object detection technology for 3D object detection. Subsequent studies converted the point cloud into voxels [7–9]. The development of sparse convolution greatly accelerated the speed of 3D convolution, and the problem of addressing the sparsity of the point cloud was solved more effectively. Thus, the voxel-based 3D object detection technology has developed rapidly, and it continues to be extensively researched. However, regardless of multi-view or voxel, a quantisation error inevitably occurs in the conversion process, which fundamentally limits the performance of these methods. With the development of PointNet [10] and PointNet++ [11], researchers have continuously extracted features directly from the original point cloud. These studies used set abstraction (SA), which involves sampling, grouping and multilayer perceptron (MLP) learning of point-wise features, in addition to using a symmetric function (max-pooling) to aggregate the features. Although the point-based method can preserve the original information of the point to a significant extent, it is still limited by the high computational and memory cost.

The attribution maps proposed by Schinagl et al. [12] make it possible to visualise the importance of each point to the prediction results. The colour distribution of the point cloud in Figure 1 shows that the foreground points play a decisive role in the target detection result, whereas the contribution of the surrounding point cloud is of lesser importance.

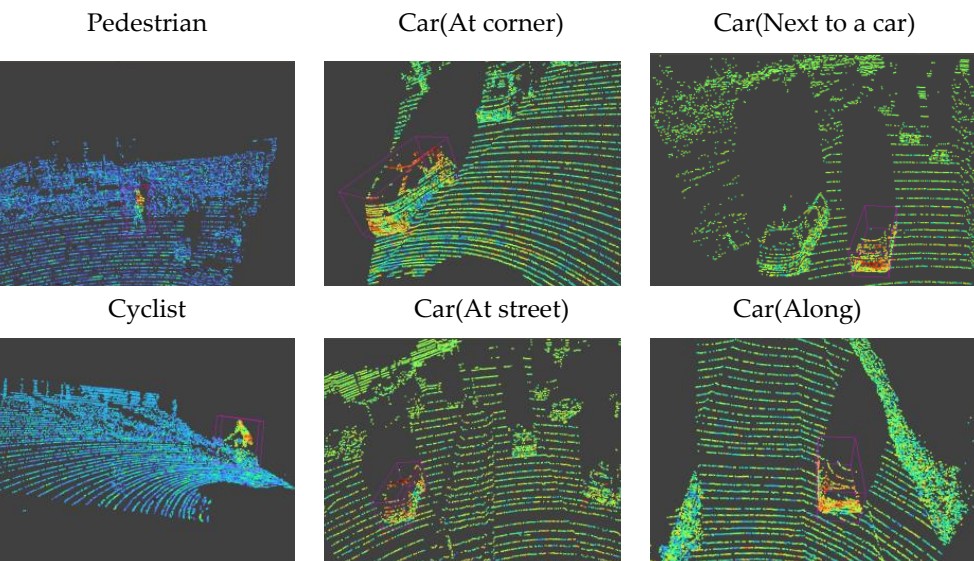

**Figure 1.** Attribution map examples for 3DSSD [13] detections on KITTI. Warmer colours (turbo colormap) denote higher contribution of a point to this detection.

In this study, we investigated point-based frameworks and explored the currently popular strategies for optimising point cloud downsampling, which is a major phase in SA. The experiments showed that the efficiency of existing heuristic sampling methods such as FPS is unsatisfactory. Before the final bounding box regression, visualisation of the sampling points revealed that the points of interest of distant sparse targets were missed before feature extraction. The dense point cloud in the immediate vicinity comprised a concentration of a large number of useless sampling points. Zhang et al. [14] proposed an instance-aware method to distinguish the points in the foreground from those in the background, but this method requires accurate extraction of the semantic information

from the point clouds, which is limited if tasks have more categories of objects. Semantics-augmented set abstraction (SASA) [15] is simple and effective, and the foreground points can be preserved as much as possible by using the weighted Euclidean distance of the foreground points. In the process of downsampling from 4096 to 1024 points, the original F-FPS technique could retain only 9.09% of the foreground points, whereas SASA could maintain a foreground rate of approximately 35.23% through the semantic-guided sampling module. In addition, almost all foreground points in the previous stage were retained in this stage, which contributed significantly to improving the candidate point generation accuracy and detection efficiency. This method generates its candidate points using VoteNet [16] (a Hough voting network).

Classifying point clouds also consumes computing resources compared to using FPS which samples points equally. This study attempted to achieve a balance between effectively distinguishing foreground and background points and efficient computation. In 3DSSD, many negative points are discarded during downsampling, which supports regression but is harmful for classification. Therefore, D-FPS needs to be additionally used to increase the negative points to provide semantic information for the candidate points. The classification accuracy can be improved by attaching sufficient semantic information to the candidate points in the feature extraction stage. Thus, the algorithm of a feature extraction method will be highly efficient if it is used to segment the category of the point cloud and attach rich global semantic information to be applied in the candidate generation (CG) stage. The attention mechanism in self-attention can adaptively aggregate the global features and achieve a similar feature enhancement effect by appending to the source features.

Traffic scenes have strong semantic correlations, such as the correlation between vehicles and road surfaces, driving directions while maintaining the same lane, and correlation between pedestrians and pedestrian crossings. The self-attention mechanism [17] captures long-range interactions and provides guidance for the selection of key points. It uses its contextual information for the classification task of the CG layer to create a positive effect. However, the self-attention mechanism is computationally expensive, which limits its use for complex datasets. Hence, a computationally cost-effective alternative that can simplify self-attention is needed.

Motivated by the above analysis, we propose architecture based on 3DSSD, namely AGS-SSD. We applied the foreground point semantic segmentation module proposed for SASA and added the point external attention (PEA) module to the SA of the PointNet++ architecture along with A-FPS (attention-guided farthest point sampling). Specifically, we used the external attention mechanism to learn the key points obtained by S-FPS and calculated the global pairwise interactions of these points. The attention map module obtained in the process was used to obtain the attention scores for A-FPS, in which the Euclidean distance was reweighted and the number of foreground points in the sampling points output by the A-FPS layer was increased. The semantic points output by the SA layer contain richer semantic information. When these points are supplied to the CG layer, the PSA is used to extract the contextual information again and provide it for the regression and classification tasks of the bounding box. Owing to its global features, the proposed approach can improve the efficiency and accuracy of 3D object detection, and enhance the detection effect of distant sparse targets.

The following are the key contributions of this study:

1.  This study identified and addressed the main problems in the point cloud downsampling method used in the existing point-based 3D object detection process, and developed an efficient and accurate 3D object detection framework based on 3DSSD, which contains two new modules: PEA and attention-guided sampling (AGS).
2.  AGS-SSD is an efficient and high-performance single-stage 3D object detection framework that can detect multiple classes of objects simultaneously. Our proposed PEA utilises an external attention mechanism to extract the long-range dependencies between the point clouds, which can save computation when compared with the self-attention approach. The PEA memory unit is more suitable for a relatively single

target in traffic scenarios. Our proposed AGS uses the attention map generated by PEA to weigh the point cloud, which enables the targeting of a greater number of sampling points and retention of a greater number of foreground points.

3. Experimental investigations on the KITTI dataset revealed that our method adequately outperforms 3DSSD on each object level and that its performance is similar to that of state-of-the-art methods, whereas its efficiency is only slightly lower than that of the baseline.

The remainder of this paper is organised as follows. Section 2 offers a detailed literature review and explains the core concepts of this study. Section 3 describes the proposed framework and the PEA and AGS modules. Section 4 presents the experimental verification of the proposed method. Finally, Section 5 presents the concluding remarks.

## 2. Related Work

In this section, we introduce point-based 3D target detection, followed by point downsampling and attention for point cloud tasks.

### 2.1. Point-Based 3D Object Detection

The point-based 3D object detection algorithm uses the original point cloud to extract the point-level features. PointRCNN [18] is the first 3D object detection algorithm to completely adopt the original point cloud. This method uses PointNet++ as the backbone for feature extraction, performs the RPN operation based on the obtained original point cloud features and adopts the standard two-stage object detection architecture. In PointGNN [19], a graph neural network is used to build a graph model based on the pre-set distance threshold; then, each vertex is updated to obtain the information on the neighbourhood points, and finally, multiple vertices are integrated to output the 3D bounding box. Point-based detection can retain the structural information of the original point to the greatest extent, and the detection accuracy has a high upper limit. However, owing to the need for expensive sampling and feature extraction, the training and reasoning time is longer than that of the voxel-based method.

### 2.2. Point Downsampling

Point cloud downsampling is a fundamental step in most point-based neural architectures, and is commonly used to refine the raw input and improve the computational efficiency for multiple downstream tasks. Recent studies have explored advanced and complex sampling schemes [20,21]. Nonetheless, despite significant progress in point cloud sampling, these methods are task-independent and fairly common, lacking knowledge of the important features that may be required for a particular task. Zhou et al. [22] builds a dense-to-sparse projection field (DBAF), obtain coordinates of keypoints and through jointly predicting confidence maps and 2D local offset fields. It provides a new approach for instance-aware downsampling.

In relation to point cloud sampling, farthest point sampling (FPS) is widely used in many models to handle the downsampling issue encountered in the use of point clouds. However, FPS has many shortcomings such as sensitivity to outliers, poor sampling effect on small samples and poor sampling effect of sparse targets. Yang et al. [23] proposed point attention transformers and the Gumbel subset sampling module, removed FPS from PointNet++ to calculate the importance of each point using Gumbel Softmax, and then selected the downsampling points according to this probability. Wang et al. [24] proposed a novel sampling method named LSNet, which can sample important points using deep learning and produces superior results regarding the number of sampling points. Yang et al. [13] proposed 3DSSD by introducing feature-FPS and fusion sampling to replace the Euclidean distance in FPS with the distance in the feature space to remove a greater number of negative points in the background. Zhang et al. [14] proposed IA-SSD, which utilises both class-aware and centroid-aware sampling strategies to retain the pre-attractions during the sampling process. Contextual centroid perception (similar to VoteNet centre-point voting)

has been proposed to regress the centre by using meaningful contextual information around the bounding box.

## 2.3. Attention Mechanism for 3D Object Detection

Owing to the disordered state of point clouds, which are essentially collections of points irregularly embedded in the metric space, the attention mechanism is well adapted for point cloud processing.

Point cloud encoders based on the attention mechanism usually perform dense prediction tasks such as objection detection and semantic segmentation. Xie et al. [25] proposed MLCV, which exploits the multi-level context of VoteNet to improve the detection performance in indoor scenes by encoding the contextual information. Specifically, the self-attention mechanism is used to strengthen the corresponding feature representations by capturing the relations within the point patches and vote clusters. Pan et al. [26] proposed Pointformer, which follows the U-Net architecture, wherein the following transformer-based blocks are proposed: local transformer (LT), local-global transformer (LGT) and global transformer (GT); this structure encodes long-range dependencies to enhance the feature representation and enhance the performance on both indoor and outdoor datasets. By following the DETR [27] framework, Nguyen et al. [28] designed Box-attention, which also serves as a multihead attention approach used to focus on the box of interest in the image feature map. To this end, it samples a grid within each box and calculates the attention weights of the sampled features from the grid structure, which makes it easy to generalise the module for 2D or 3D object detection and instance segmentation. In each attention computation head, a box of interest is generated by predicting a geometric transformation of a predefined reference window (e.g., translation, zoom and rotation).

## 3. Method

As shown in Figure 1, the AGS-SSD framework employs a PEA module to produce more powerful and robust global context representations from the features extracted by the backbone network. The AGS-SSD also replaces the D-FPS (Euclidean distance farthest point sampling) in the SA layer with A-FPS (attention-guided FPS), which aims to achieve higher foreground point recall. In brief, we make full use of the point cloud context feature extracted from PEA to boost the limited precision of 3D object detection caused by foreground point sampling loss and the lack of semantic information.

## 3.1. Overall Structure

The overall architecture of AGS-SSD is shown in Figure 2. The framework contains three parts: a point feature extraction backbone, CG layer and prediction head. The input to the network is the original point cloud, and the backbone is based on PointNet++, which contains three SA layers. One SA layer is divided into three parts—downsampling, grouping and MLP layers—which extract the global and local features of the points of interest at different scales. By referring to 3DSSD, our backbone retains only the encoder layers and abandons the decoder layers, which are the expensive FP up-sampling layers. The extracted features are provided to the CG layer, which contains a vote head for processing. The vote head contains a simplified version of PointNet; it uses MLP to return the offset of the point of interest, which provides the centre of the object and the centre of its features. Finally, the CG layer sends the resulting object centre and candidate box to the classification and regression network using the fully connected layer to obtain the final result.

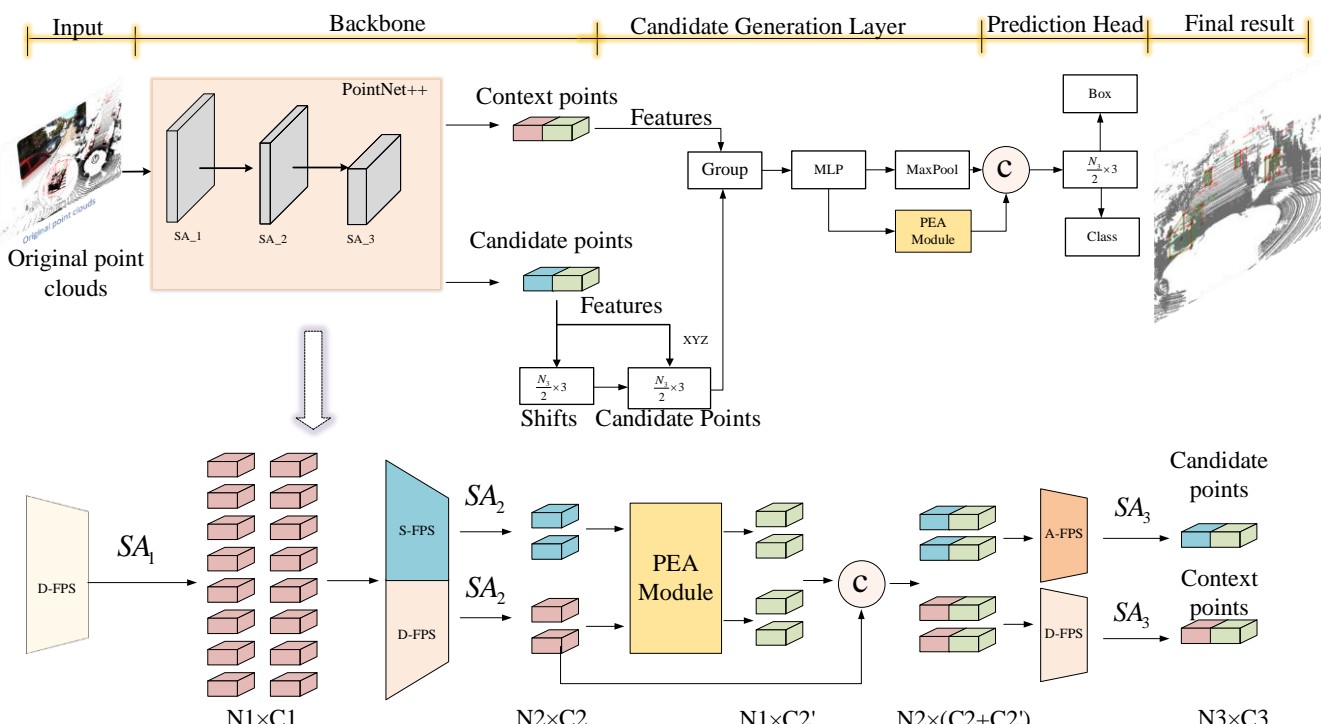

**Figure 2.** Illustration of the entire AGS-SSD framework. First, the network performs feature extraction using three SA layers. Second, the PEA module is inserted between $SA_2$ and $SA_3$ to extract the contextual information on the features extracted by $SA_2$. The PEA module is also added to the CG layer to shift the sampled points to the corresponding instance centres. Third, the A-FPS module samples the key point set K based on the point coordinates X and attention scores P. Finally, the prediction head gives the class score and regression bounding box prediction. In the lower part of the image, the blocks output by SA represent the ratio of the cloud of the downsampling points, i.e., 16:4:2:1.

The three SA encoding modules can be classified as $SA_1$, $SA_2$ and $SA_3$. The corresponding numbers of subsampled layers are level1, level2 and level3, which are used to extract the point-wise feature vectors. In contrast to the original 3DSSD implementation, we refer to SASA and replace the F-FPS of the 3DSSD with S-FPS at $SA_2$. Compared with D-FPS (Euclidean distance-based FPS), the S-FPS adds a two-layer MLP structure as a simple foreground point segmentation module. The number of downsampled point clouds is $[16,383 \rightarrow 4096 \rightarrow (512,512) \rightarrow (256,256)]$. The output (512,512,512) of the level2 stage passes through the PSA module and outputs the global context features; it then performs splicing with the point-wise feature vectors output from the level2 stage. The sampling points of the S-FPS module are sent to the AGS module for attention-weighted sampling. The input 512 point clouds are weighted by the attention weights and distance weights to obtain the score, and the top 256 points are removed and sent to VoteNet as the candidate generating points. VoteNet takes the input candidates as the central points, combines the semantic features obtained from the D-FPS sampling and extracts the features using the SA layer containing the PEA module. These features are sent to the prediction head for regression and classification.

### 3.2. PEA Module

Attention mechanisms are often used to extract the local–global relationships and to reinforce key features and suppress useless features. The attention mechanism is widely employed in 2D image target detection. Self-attention modelling helps achieve state-of-the-art results in machine translation, and self-attention is commonly combined with

convolution in natural language processing, image recognition, two-dimensional object detection, segmentation, reinforcement and learning.

Inspired by the external attention proposed by Guo et al. [29], the scheme proposed in this paper applies the PEA module, which is less computationally intensive when compared with the self-attention mechanism, yet it captures the relationship between the samples. Such a structure would be considerably effective in a scenario wherein the vehicle, pedestrian, bicycle and other targets have an obvious semantic correlation with the pavement and sidewalk background.

Our PEA module is shown in Figure 3b. Compared with the point cloud self-attention module [30] in Figure 3a, PEA adopts the external attention method. As shown in the figure, the PEA module first calculates the attention map by calculating the similarity between the self-query vector and the external learnable key memory $M_k$. It then multiplies the attention graph with another memory unit $M_v$ to obtain the feature graph. An optional training module is available for use when the sampling points are required to be focused within the former attractions as much as possible.

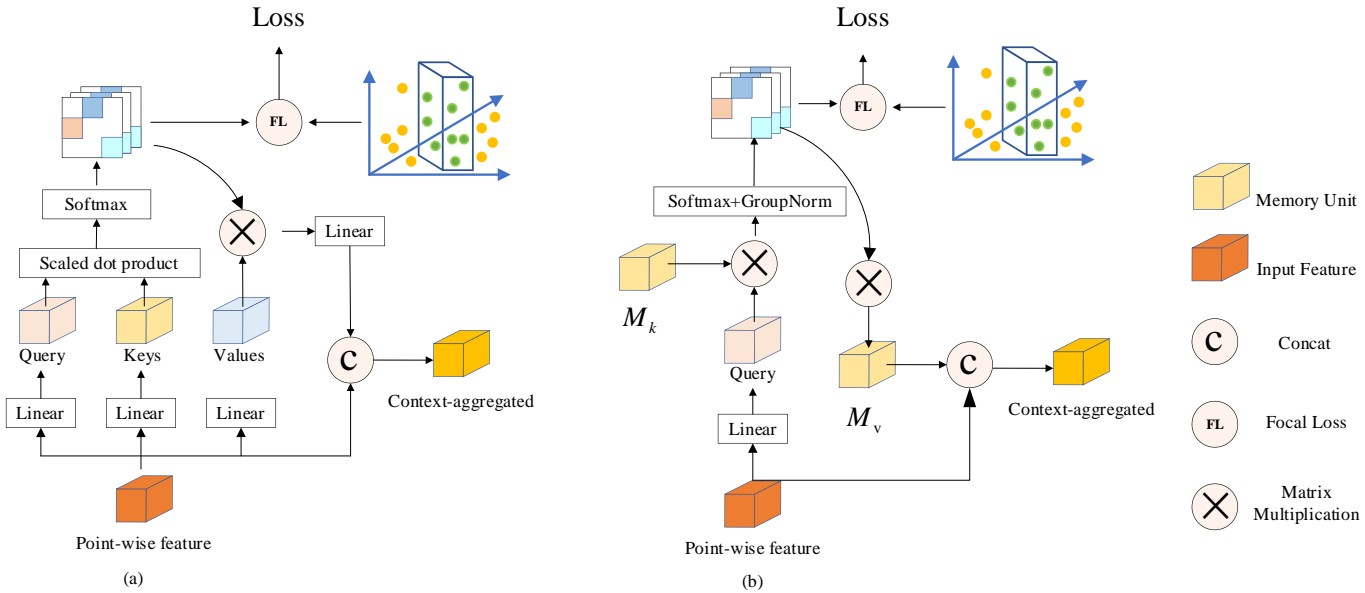

**Figure 3.** PEA module. (**a**) Point self-attention module; (**b**) point external attention module.

The matrix M can be shared to learn the macrorelationship between the overall data samples. Similar to the shared MLP operation of PointNet, the memory units calculate the attention for each point and establish the microrelationships (relationships between the self-attention pairs).

**Complexity**. The complexity of the self-attention calculation is $O(N^2d)$; the complexity of the external attention module is $O(Nsd)$, where $s$ is the dimension of the memory block. Based on the number of input point clouds, $N$, in the square relationship, a smaller value of $s$ was selected in this study to achieve a similar effect as SA. In this study, $s$ was set to 64, and the complexity of PEA was only 1/4 or 1/8 of that of SA for input $N$ = 512 or 256. The inherent sparsity of the point cloud and efficient pairwise computing based on matrix multiplication make PSA a feasible feature extractor in the current 3D detection architecture.

**Multihead attention.** Point cloud self-attention module employs the multihead self-attention mechanism. In contrast to single-head attention, multihead attention can capture different relations between the tokens. Inspired by this idea, a multihead external attention mechanism was designed in this study. Specifically, the input feature first passes through a linear layer and is then divided into H heads; the matrix after the rank is $M_k$ transformed

and then normalised, followed by $M_v$ transformation. Finally, feature splicing is performed; the formula can be expressed as follows:

$$h_i = PEA(F_i, M_k, M_v), \tag{1}$$

$$F_{out} = MultiHEAD(F, M_k, M_v) = Concat(h_1, \ldots, h_H)L_{reduce}, \tag{2}$$

where $h_i$ is the $i$-th head, $H$ is the number of heads and $L_{reduce}$ is a linear transformation matrix, which reduces the dimensions of the output features to make it consistent with the input features. $M_k \in R^{S \times d}$ and $M_v \in R^{d \times S}$ are the shared memory units for the various heads.

Unlike SA multihead mechanism, EA multihead mechanism uses a shared connection layer. With this structure, the number of parameters can be reduced. In the shared memory unit, $H$ is multiplied by $k$ and $S$ is divided by $k$. The specific structure is shown in Figure 4.

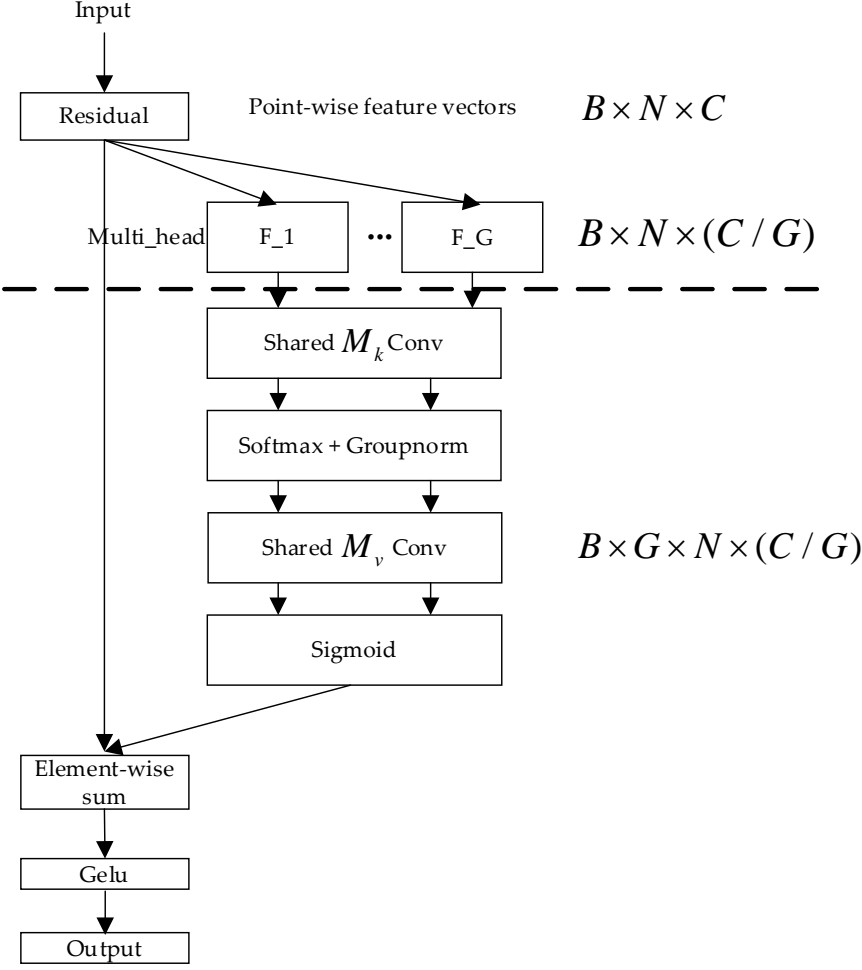

**Figure 4.** Multihead external attention.

### 3.3. Attention-Guided Sampling

Inspired by SASA, the MLP prediction of the former attractions and background points is of great help in improving the sampling efficiency. In SASA, a lightweight point segmentation is embedded to improve the identification of the local semantics. Specially, it is a simple 2-layer MLP and classifies the input points into foreground and background points. The point segmentation module denotes the $C_k$-dimension point-wise features fed to the $k$-th SA layer as $\{f_1^{(l_k)}, f_2^{(l_k)}, \ldots f_{Nk}^{(l_k)}\}$. The foreground score $p \in [0, 1]$ for each point is calculated as

$$p_i = \sigma(MLP_k(f_i^{(l_k)})), \tag{3}$$

where $MLP_k$ is a 2-layer MLP, which maps the input features to the foreground scores and $\sigma\left(\cdot\right)$ is a sigmoid function.

Unlike SASA, AGS-SSD employs the attention map generated by the attention mechanism. The map is provided to the sigmoid network to obtain the confidence of the region-of-interest point cloud in [0, 1], which is added to the sampling distance as the score. As shown, the semantic labels are added according to the point cloud inside the bound box in the ground truth.

The advantage is that the attention map can continue to operate with attention scores to predict the properties of the point clouds to obtain the contextual information and provide the precise candidate points for the CG layer. Given the point attention map extracted by the previous PEA module and point coordinates from the input, the process of our proposed A-FPS is described in Algorithm 1. The main purpose is to select a greater number of foreground points by giving precedence to the points with higher attention scores. The overall procedure of FPS remains unchanged; we rectify the sampling metric, which is the distance to the already-sampled points, using the point attention scores. Specifically, given the coordinates $\{x_1, \ldots, x_N\}$ and attention scores $\{A_1, \ldots, A_N\}$ of the input points, a distance array $\{d_1, \ldots, d_N\}$ maintains the shortest distance from the $i$-th point to the already selected key points. In each round of selection, we add the point with the highest attention-weighted distance $D_i$ to the key point set. $D_i$ is computed as

$$D_i = (A_i)^\gamma \cdot d_i \qquad (4)$$

where $\gamma$ is the balance factor controlling the contribution of the attention scores.

---

**Algorithm 1:** Attention-guided farthest point sampling algorithm. $N$ is the number of input points and $M$ is the number of output points sampled by the algorithm

---

**Input:**
Coordinates $X = \{x_1, \ldots, x_N\} \in \mathbb{R}^{N \times 3}$
Attention scores $A = \{A_1, \ldots, A_N\} \in \mathbb{R}^N$
**Output:**
Sampled key point set $K = \{k_1, \ldots, k_M\} \in \mathbb{R}^{M \times 3}$

1.    Initialise an empty sampling point set $K$
2.    Initialise a distance array d of length $N$ with all $+\infty$
3.    Initialise a visit array $v$ of length $N$ with all zeros
4.    **for** i = 1 to $M$ **do**
5.      **if** i = 1 **then**
6.        $k_i = argmax(A)$
7.      **else**
8.            $D = \{(A_k)^\gamma \cdot d_k | v_k = 0\}$
9.            $k_i = \text{argmax}(D)$
10.     **end if**
11.       add $k_i$ to K, $v_{ki} = 1$
12.       for $j = 1$ to $N$ do
13.         $d_j = \min(dj, \left|\left|x_j - x_{ki}\right|\right|)$
14.     **end for**
15.   **end for**
16.   **Return K**

---

The role of AGS is shown in Figure 5. Suppose $P_1$ is the starting point. If FPS is used, select $P_2$ such that $D\left(P_1, P_2\right) > D\left(P_1, P_3\right)$. However, the attention scores of $P_2$ and $P_3$ are related as $a_3 > a_2$ for $P_3$ as the foreground point. Hence, $a_2{}^\gamma \times D\left(P_1, P_2\right) < a_3{}^\gamma \times D\left(P_1, P_3\right)$ is obtained using A-FPS, thus leading to $P_3$ being selected instead of $P_2$.

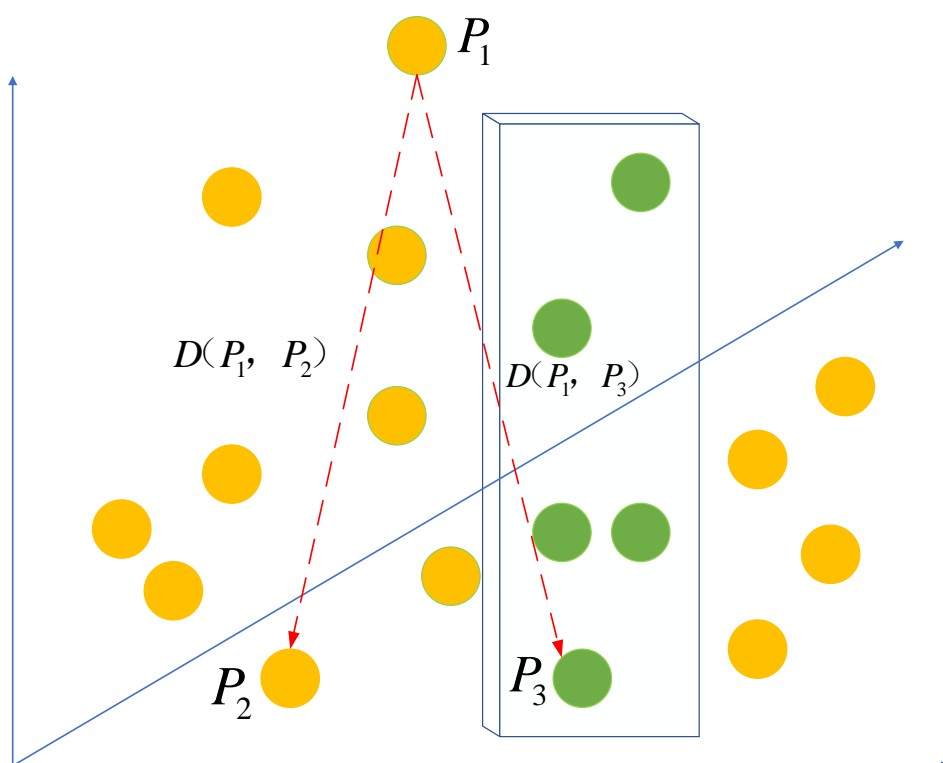

**Figure 5.** 2D illustration of A-FPS.

## 4. Experiments

### 4.1. Dataset

We validated our AGS on the popular KITTI dataset. This dataset is a widely used public dataset in the field of 3D object detection and is a prevailing benchmark. It contains 7481 LiDAR point clouds as well as the corresponding images with calibrated 3D bounding boxes for training. The instances are divided into three categories: cars, pedestrians and cyclists. The dataset also contains 7518 unlabelled samples for testing. The results of the test can be obtained only by submitting the inferences to the official server dedicated to KITTI.

By adopting the OpenPCDet setting, the 7481 training samples were divided into train split (3712 samples) and *val* split (3769 samples) in this study. All experimental models were trained on the train split and tested on the val split in the evaluation stage. Following the training protocol suggested for PV-RCNN [31], the model was trained with 80% of all the data in the training set and the remaining 20% of the data were used for validation.

Because the dataset annotates only the objects that are visible in the image, the point cloud was processed within the field of view of the image. We performed all experiments on three objects: cars, pedestrians and cyclists. Three difficulty levels (easy, medium and hard) were considered depending on the size of the 3D object, occlusion level and truncation. For training purposes, samples that did not contain objects of interest were removed.

Average precision (AP) is the main evaluation metric for object detection models. The evaluations of both the test set and val set used 40 recall positions, represented as AP_40, which is more scientific than the assessment method based on 11 recall positions.

### 4.2. Implementation Details

**Baseline.** 3DSSD was selected as the baseline, as it has an efficient 3D single-stage detector. Our experimental model for evaluation was built on OpenPCDet, a widely used and clean 3D object detection framework. It supports most 3D object detection algorithms, including single-stage and two-stage, voxel-based and point cloud-based, and offers a variety of detection heads.

**Traning.** The reproduction of other methods was based on the official configuration provided by OpenPCDet. As in the case of the baseline, all networks were trained using the ADAM optimiser with an initial learning rate of 0.001 and batch size of 16 equally distributed across 4 TESLA V100 GPUs. All models were trained for 80 epochs. To align the network input, each point cloud contained 16,384 randomly selected points. If the number of points in one scene was fewer than 16,384, the points were randomly repeated to obtain 16,384 points.

**Data Enhancement.** To avoid overfitting, we employed manifold data augmentation strategies on the KITTI dataset. First, we randomly added the foreground instances and their interior points from other scenes to the current point cloud by using the same mixing strategy $\Delta\theta_1 \in [-\pi/4, +\pi/4]$. Next, we rotated each bounding box by following a uniform distribution and added a random translation $[\Delta x, \Delta y, \Delta z]$. Third, we applied random scaling with a scale from [0.9, 1.1] and each point cloud was randomly flipped along the $x$-axis.

*4.3. Results*

As indicated in Table 1, our AGS-SSD outperformed the 3DSSD baseline by a large margin in all aspects, thus establishing itself as a state-of-the-art point-based single-stage detector.

**Table 1.** Quantitative detection performance achieved by different approaches on the KITTI *test* set. All results were evaluated based on mean average precision with 40 recall positions via the official KITTI evaluation server. The results of our AGS-SSD are shown in bold, and the best results are bold, and the second best results are underlined.

| Method | Type | Car 3D Detection (IoU = 0.7) | | | Ped.3D Detection (IoU = 0.7) | | | Cyc.3D Detection (IoU = 0.7) | | | Speed |
|---|---|---|---|---|---|---|---|---|---|---|---|
| | | Easy | Mod. | Hard. | Easy | Mod. | Hard. | Easy | Mod. | Hard. | |
| SECOND [7] | 1-stage | 84.65 | 75.96 | 68.71 | 45.31 | 35.52 | 33.14 | 75.83 | 60.82 | 53.67 | 20 |
| PV-RCNN [31] | 2-stage | <u>90.25</u> | 81.43 | 76.82 | 52.17 | 43.29 | <u>40.29</u> | 78.60 | **63.71** | **57.65** | 12.5 |
| Voxel R-CNN [9] | 2-stage | **90.90** | <u>81.62</u> | <u>77.06</u> | - | - | - | - | - | - | 25 |
| PointPillars [32] | 1-stage | 82.58 | 74.31 | 68.99 | 51.45 | 41.92 | 38.89 | 77.10 | 58.65 | 51.92 | 42.4 |
| PointRCNN [18] | 2-stage | 86.96 | 75.64 | 70.70 | 47.98 | 39.37 | 36.01 | 74.96 | 58.82 | 52.53 | 10 |
| PointGNN [19] | 1-stage | 88.33 | 79.47 | 72.29 | 51.92 | <u>43.77</u> | 40.14 | 78.60 | 63.48 | <u>57.08</u> | 1.6 |
| 3D IoU Net [33] | 1-stage | 86.16 | 76.50 | 71.39 | - | - | - | - | - | - | 12.5 |
| Fast Point R-CNN [34] | 2-stage | 85.29 | 77.40 | 70.24 | - | - | - | - | - | - | 16.7 |
| Part-$A^2$ [35] | 2-stage | 87.81 | 78.49 | 73.51 | <u>53.10</u> | 43.35 | 40.06 | **79.17** | <u>63.52</u> | 56.93 | 12.5 |
| TANet [36] | 2-stage | 84.39 | 75.94 | 68.82 | 53.72 | **44.34** | **40.49** | 75.70 | 59.44 | 52.53 | 28.5 |
| 3DSSD (Official) [13] | 1-stage | 88.36 | 79.57 | 74.55 | | | | | | | 25 |
| 3DSSD (OpenPCDet) | 1-stage | 87.91 | 79.55 | 74.71 | 35.03 | 27.76 | 26.08 | 66.69 | 59.00 | 55.69 | 26 |
| SASA [15] | 1-stage | 88.76 | **82.16** | **77.16** | - | - | - | - | - | - | 27 |
| AGS-SSD (Ours) | 1-stage | 88.38 | 81.02 | 76.45 | 46.10 | 38.53 | 35.40 | 77.40 | 62.15 | 56.14 | 24 |

Regarding the main metric, i.e., AP_40 on objects in the car category at the "moderate" level, our method outperformed 3DSSD and PointRCNN by 1.45% and 5.38%, respectively. We should note that many methods evaluate only one class on the KITTI test set, mainly the "car" class. This is mainly because of the following considerations: first, the samples in KITTI have a highly unbalanced distribution, with the proportion of cars and pedestrians approximating 10:1. However, the test set contains a small number of images of pedestrians and bicycles, which easily leads to huge fluctuations in the test results, making the results meaningless. The second reason is that some algorithms adopt a task-driven detection strategy, and corresponding optimisations are conducted in the stages of setting the target anchor frame and constructing the loss function. This makes them unsuitable for multi-category target detection. However, this does not mean that the methods cannot be applied to other categories. We also mention the detailed results of the 3DSSD test, as it is important to evaluate the performance of the baseline. 3DSSD does not provide official replication implementations or pre-training models for pedestrian and cyclist data. Because our AGS-SSD was built on the basis of OpenPCDet, we used a duplicate version of OpenPCDet for fair comparison.

The AP_40 scores of cars in the test set at the easy, moderate and hard levels, obtained by AGS-SSD, were 88.38%, 81.02% and 76.45%, respectively. The proposed method outperformed most of the recent detectors, e.g., PointRCNN, 3DSSD and SECOND. Its detection accuracy slightly inferior to that of PV-RCNN and Voxel R-CNN but its inference speed is faster than that of them.

The KITTI server limits the number of test set submissions. Hence, to compare the performance more comprehensively, the method proposed in this paper was compared with the current state-of-the-art method. The AP for 3D object detection by our AGS-SSD method on the KITTI dataset is presented in Table 2. The results in this table depict the AP_40 value calculated with the same (IoU) setting on the test set. As shown in the table, our AGS-SSD delivers the best performance for the easy level on the car class of the val set. AGS-SSD performs slightly more accurate than 3DSSD SASA in every aspect. It is not uncommon for our algorithm to deliver far superior performance on the val set than on the *test* set. Classical algorithms such as PointRCNN have similar performance characteristics owing to the imbalance between the categories and some differences in the algorithm design. Figure 6 presents the visualization results of AGS-SSD on the KITTI dataset.

**Table 2.** Quantitative detection performance achieved by different methods on the KITTI val set.

| Method | Car 3D Detection | | | Ped. 3D Detection | | | Cyc. 3D Detection | | |
|---|---|---|---|---|---|---|---|---|---|
| | Easy | Mod. | Hard | Easy | Mod. | Hard | Easy | Mod. | Hard |
| PV-RCNN [31] | 90.25 | 81.43 | 76.82 | 52.17 | 43.29 | 40.29 | 78.60 | 63.71 | 57.65 |
| 3DSSD (OpenPCDet) | 91.04 | 82.32 | 79.81 | 59.14 | 55.19 | 50.86 | 88.05 | 69.84 | 65.41 |
| IA SSD [14] | 91.88 | 83.41 | 80.44 | 61.22 | 56.77 | 51.15 | 88.42 | 70.14 | 65.99 |
| CT3D [37] | 92.34 | 84.97 | 82.91 | 61.05 | 56.67 | 51.10 | 89.01 | 71.88 | 67.91 |
| PDV [38] | 92.56 | 85.29 | 83.05 | 66.90 | 60.80 | 55.85 | 92.72 | 74.23 | 69.60 |
| SASA [15] | 91.82 | 84.48 | 82.00 | 62.32 | 58.02 | 53.30 | 89.11 | 72.61 | 68.19 |
| AGS-SSD (car) | 92.4 | 85.28 | 82.01 | - | - | - | - | - | - |
| AGS-SSD (multi) | 92.22 | 85.08 | 82.35 | 62.55 | 59.35 | 54.16 | 90.65 | 74.19 | 69.79 |
| Improvement on 3DSSD | +1.18 | +2.76 | +2.54 | +3.41 | +4.16 | +3.30 | +2.60 | +4.35 | +4.38 |

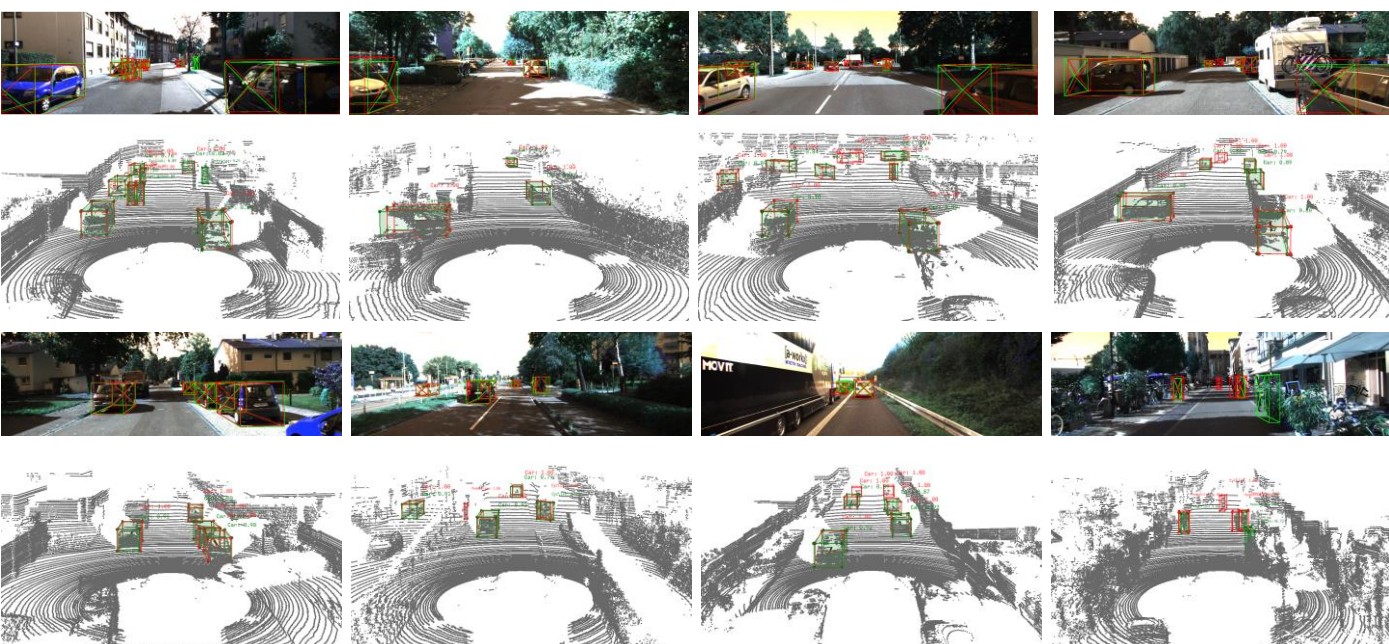

**Figure 6.** Qualitative results achieved on the KITTI val set. Red boxes represent the ground truth. Green boxes denote the prediction results. The numbers in green represent the confidence of the prediction.



Overall, the results of the model on both the val and test sets consistently demonstrate that our proposed AGS-SSD achieves the state-of-the-art AP on 3D object detection and high efficiency when compared with the baseline.

### 4.4. Ablation Experiments

**Ablation on Modules.** To further verify the effectiveness of the PEA and AGS modules, we considered the original 3DSSD SASA as the baseline. We replaced the original S-FPS in Level 3 with A-FPS and added the PEA module to extract the contextual information. The results are presented in Table 3 and show that, by adding the external attention module, the accuracy was improved by 0.44%, 1.21% and 1.27% for car, pedestrian and cyclist, respectively. The attention mechanism highlights the global contextual information to improve the regression, and weakens the effect of the futile noise to make the algorithm more robust. By adding A-FPS, we can obtain a greater number of targeted sampling points.

**Table 3.** Performance of proposed method with different configurations on KITTI val set. The results are evaluated with the average precision calculated based on 40 recall positions.

| Baseline | PEA | A-FPS | Car Mod (IoU = 0.7) | Ped Mod (IoU = 0.5) | Cyc Mod (IoU = 0.5) |
|:---:|:---:|:---:|:---:|:---:|:---:|
| √ | | | 84.48 | 58.02 | 72.61 |
| √ | √ | | 84.92 | 59.23 | 73.88 |
| √ | √ | √ | 85.08 | 59.35 | 74.19 |

**Sampling point recall.** The detection performance and point recall, which is the ratio of the number of ground truth (GT) boxes with at least one internal sample point to all GT boxes for different samples, are compared in Table 4. The algorithm is based on the 3DSSD baseline. We adjusted only the point sampling policy; the other model settings remained unchanged. The results showed that our A-FPS outperformed the F-FPS used in the 3DSSD baseline by up to 2.28%. Furthermore, the candidates sampled by our A-FPS could "hit" 0.24% additional real boxes when compared with S-FPS. As can be seen from the visualisation results in Figure 7, FPS was used in Layer 1 and Layer 2 to obtain a uniform sampling point, whereas A-FPS was used in Layer 3 to ensure a greater concentration of the foreground points. Finally, the 256 sampling points output in the CG layer were almost entirely concentrated around the instances.

**Table 4.** Analysis of point sampling by layer on 3DSSD, evaluated on the car class of KITTI val split. The level 2 and level 3 SA layers exploit the fusion sampling strategy to individually sample half the key points with two different sampling algorithms.

| Method | SA Layer Total Points | $SA_1$ 4096 | $SA_2$ 512 | $SA_2$ 512 | $SA_3$ 256 | $SA_3$ 256 |
|:---:|:---:|:---:|:---:|:---:|:---:|:---:|
| 3DSSD | Sampling Method: Foreground Rate: Point Recall: | FPS 4.4 98.35 | F-FPS 9.09 97.75 | FPS 3.2 96.73 | F-FPS 8.7 96.65 | FPS 2.75 91.58 |
| 3DSSD + SASA | Sampling Method: Foreground Rate: Point Recall: | FPS 4.4 98.35 | S-FPS 35.23 97.87 | FPS 3.2 96.73 | S-FPS 31.24 97.65 | FPS 2.75 91.58 |
| 3DSSD + AGS | Sampling Method: Foreground Rate: Point Recall: | FPS 4.4 98.35 | S-FPS 35.23 97.87 | FPS 3.2 96.73 | A-FPS 33.52 97.89 | FPS 2.75 91.58 |
| AGS Improvement on 3DSSD | Foreground Rate: Point Recall: | - - | - - | - - | +2.28 +0.24 | - - |

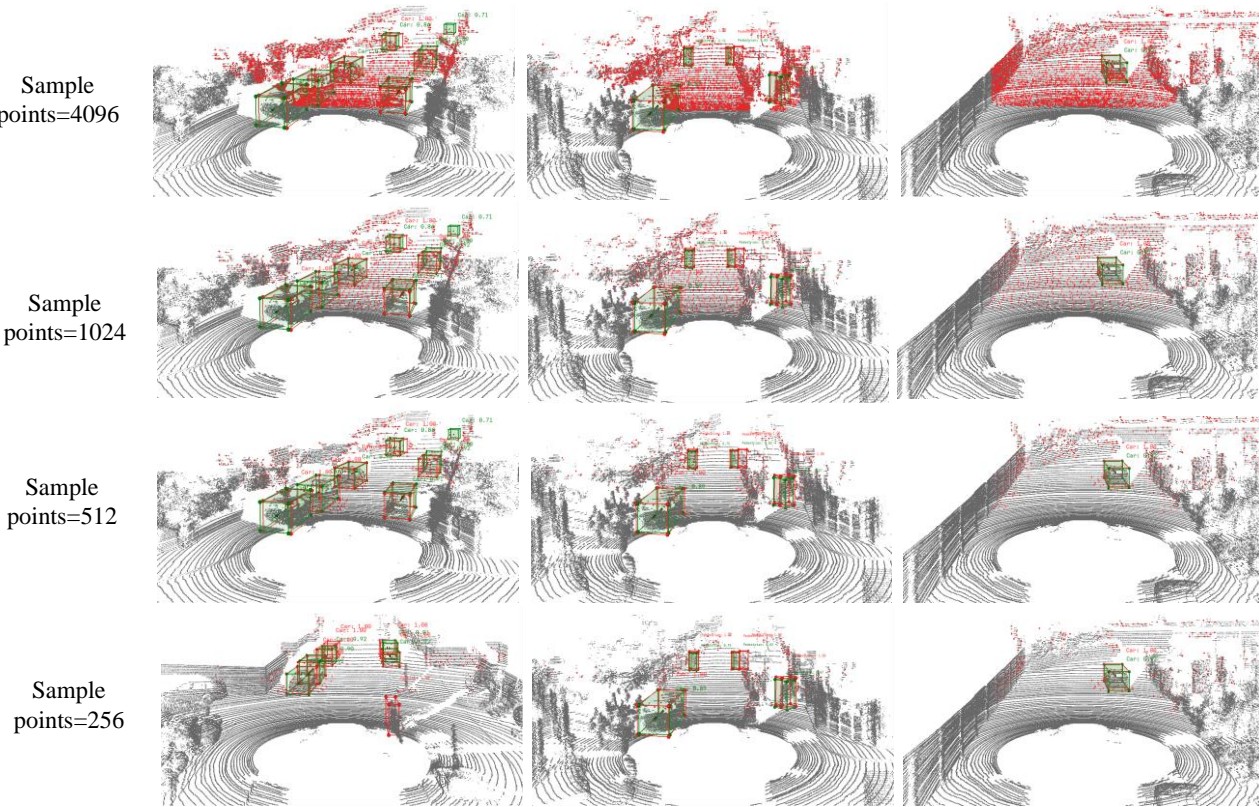

**Figure 7.** Qualitative visualisation of the downsampled point clouds achieved at different sampling stages.

**Effectiveness of hyper-parameter:** The effectiveness of balance factor $\gamma$ has been compared in Table 5. The results show that only the appropriate $\gamma$ can make the detector play the best accuracy. It cannot be too small, otherwise the attentional information will be difficult to function. It also cannot be too large, otherwise the sampling area will focus too much on some salient regions, which is unfavourable for detecting sparse and small objects at long distances. When $\gamma = 1$, we have a satisfactory result.

**Table 5.** Performance comparison with different balance factor settings on A-FPS.

|                | Car Mod (IoU = 0.7) | Ped Mod (IoU = 0.5) | Cyc Mod (IoU = 0.5) |
|----------------|---------------------|---------------------|---------------------|
| $\gamma = 0.01$ | 79.69               | 53.64               | 65.16               |
| $\gamma = 0.5$  | 82.36               | 57.16               | 72.3                |
| $\gamma = 1$    | 85.08               | 59.35               | 74.19               |
| $\gamma = 10$   | 83.87               | 51.79               | 67.08               |

## 5. Conclusions

In this article, we proposed AGS-SSD, which employs a PEA module and an A-FPS strategy to alleviate the problem of limited detection accuracy resulting from the point cloud imbalance between the foreground and background points. The PEA module uses the external attention mechanism by introducing a memory unit. Compared with self-attention, the memory usage and computational cost of external attention is lower, and the global memory stored by the memory unit is more suitable for situations where the combination of semantic features such as traffic scenes does not change frequently. Therefore, the context extracted by the external attention can provide more accurate semantic information to the improved VoteNet and increase the accuracy of centre-point prediction. AGS reorders the sampling points according to the attention weight, and the final results consider the sparse

foreground points. The advantage of VoteNet in comparison with a simple segmentation network is that it can extract deeper features of the target, and the extracted segmentation results are more accurate. At the same time, the extracted features can also be used for centre-point prediction in the CG layer. The experimental results indicate that AGS-SSD outperforms the original 3DSSD on the KITTI benchmark. We consider the test on the KITTI *test* set to be a long way from attaining its full potential, because all our results on the val test have reached the state-of-the-art level. In the next step, we need to continue to study ways in which to optimise the attention mechanism based on the point cloud, improve its efficiency and endeavour to experiment on more complex datasets such as Once.

**Author Contributions:** Conceptualisation and methodology, H.Q.; validation, H.Q.; writing—original draft preparation, H.Q.; writing—review and editing, H.Q. and P.W.; supervision, S.S.; funding acquisition and resources, B.S. All authors have read and agreed to the published version of the manuscript.

**Funding:** This work was supported by National Natural Science Foundation of China, grant Number 5210377; Hunan Provincial Innovation Foundation for Postgraduate CX20210020; National Defense University of Science and Technology Intramural Project 2022211.

**Data Availability Statement:** Data available in a publicly accessible repository that does not issueDOIs. Publicly available datasets were analyzed in this study. This data can be found here: [http://www.cvlibs.net/datasets/kitti/index.php, accessed on 10 July 2022].

**Conflicts of Interest:** The authors declare no conflict of interest.

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
