# Peer review of "AGS-SSD: Attention-Guided Sampling for 3D Single-Stage Detector"

_electronics, doi:10.3390/electronics11142268_

Round 1
Reviewer 1 Report
The paper tackles the task of 3D object detection by proposing a network named AGS-SSD. The method has some novelty and the results look promising. However, there are some issues that should be addressed:
(1) Many references are with incorrect form. Please improve them.
(2) I am wondering whether the method directly applies pointnet++ without modification. If yes, the bottom part in Fig. 2 will be redundant.
(3) What's the main advantage of PEA over point self-attention? Is the lower complexity the main superority?
(4) Some relevant works are missed. Towards a weakly supervised framework for 3d point cloud object detection and annotation also focuses on 3D object detection. The attention-guided sampling technique is very similar the feature upsampling in Differentiable multi-granularity human representation learning for instance-aware human semantic parsing, which should be discussed.
(5) How is the hyper-parameter \gamma (Eq.4) determined?
Reviewer 2 Report
The article deals with the detection of 3D objects using LiDAR. The essence lies in the appropriate processing of the cloud of points. The development of suitable algorithms for point cloud processing is very important for practical applications of this method of 3D object detection. The main contribution of this work is the creation of the AGS-SSD methodology, which contains two modules that examine the correlation between data and the necessary information is obtained by appropriate sampling.
The issue of 3D object recognition is important for several areas of science and technology. These are, for example, applications in automotive, where this technology could be applied to recognize objects on the road. Another very important area is applications in service and industrial robotics applications that need appropriate machine vision technology. Therefore, this article is very important not only for scientific contributions but also for practical applications.
In the introduction, the situation is analyzed in detail and related works in this area of research are also listed.
The methods used are described in detail in the next section.
An essential and important part of the article are the experiments that confirm the correctness of the methods used to solve the given problem. The results are processed in the form of tables and would be much clearer than graphs. Please consider a better form of presentation of these results.
Comments:
There are wrong references in the article: "[Error! Reference source not found.]." it needs to be fixed.
Line 69 on page 2: "attributionmaps" - missing space
Some symbols are written using the equation editor and have a larger font than normal text (line 285, 286, 296, 297, 300, 308, 309, etc.). It needs to be matched with the template. Check the full article.
Line 300: N should be as a quantity in italic format if it is not a matrix
Line 308, 316, 317: H, k, s should be as quantity in italic format if not matrix
equation (1) hi - should be like italic style
It is necessary to check the entire article and correct the style of writing mathematical quantities and symbols in the text and also in the equations.
Round 2
Reviewer 1 Report
The revision has addressed my concerns.